# Fruit By-Products and Their Industrial Applications for Nutritional Benefits and Health Promotion: A Comprehensive Review

**Ejigayehu Teshome** [1,2], **Tilahun A. Teka** [2,*], **Ruchira Nandasiri** [3], **Jyoti Ranjan Rout** [4],
**Difo Voukang Harouna** [5], **Tessema Astatkie** [6] **and Markos Makiso Urugo** [1,2]

1 Department of Food Science and Postharvest Technology, College of Agricultural Science, Wachemo University, Hosanna 667, Ethiopia
2 Department of Postharvest Management, College of Agriculture and Veterinary Medicine, Jimma University, Jimma 378, Ethiopia
3 Department of Food and Human Nutritional Sciences, Faculty of Agriculture and Food Science, University of Manitoba, Winnipeg, MB R3T 2N2, Canada
4 School of Biological Sciences, AIPH University, Bhubaneswar 752101, Odisha, India
5 Department of Biological Sciences, Genetic, Genomic, Proteomics & Food for Nutrition Research Unit, Faculty of Sciences, The University of Maroua, Maroua 814, Cameroon
6 Faculty of Agriculture, Dalhousie University, Truro, NS B2N 5E3, Canada
* Correspondence: tilaeta@gmail.com or tilahun.abera@yahoo.com or abera.teka@ju.edu.et

**Abstract:** Fresh and processed fruits are commonly used to prepare different industrial products with superior nutritional and health-promoting properties. Currently, the demand for processed-fruit products has motivated the rapid growth of fruit-processing industries, persuading them to produce an enormous number of by-products. Furthermore, people's shifting dietary habits and lack of awareness of nutritional properties result in a large number of fruit by-products. The lack of knowledge about the value of by-products urges the exploration of proper documents that emphasize the health benefits of such products. Hence, this article was prepared by carefully reviewing the recent literature on industrial applications of fruit by-products and their nutritional and health-promoting properties. The use of fruit by-products in food industries for various purposes has been reported in the past and has been reviewed and described here. Fruit by-products are a good source of nutrients and bioactive components, including polyphenols, dietary fibers, and vitamins, implying that they could have an important role for novel, value-added functional food properties. Furthermore, fruit by-products are used as the substrate to produce organic acids, essential oils, enzymes, fuel, biodegradable packaging materials, and preservatives.

**Keywords:** biomass valorization; biotechnological techniques; food waste; fruit processing; waste utilization

## 1. Introduction

According to the Food and Agriculture Organization (FAO), there is a shift in consumers' demand from processed foods to natural foods of superior quality that meet their nutritional requirements while promoting health [1]. Fruits and vegetables are rich in nutritional value and often promote human health. This is because fruits and vegetables are packed with vitamins, antioxidants, minerals, and dietary fiber. Citrus, watermelon, banana, apple, grape, and mango are the most popular fruits produced in the world [2]. The global production statistics include 124.8 million metric tons (MMT) of citrus, 114.1 MMT of bananas, 84.6 MMT of apples, 74.5 MMT of grapes, 45.2 MMT of mangoes, and 25.4 MMT of pineapples [2].

Even though fruit production statistics show an annual increase, this increase is still insufficient to meet the consumption demand. Increasing global population, as well as a lack of efficient production and supply chains, frequently necessitates the development of new innovative technologies to meet demands [3]. According to the FAO, over 821 million

people are currently malnourished due to a scarcity of staple foods such as starchy cereals, roots, and tuber crops [2]. This creates a shift from conventional plant-based diets to other substitute food products, including the by-products of fruits and vegetables.

Furthermore, a recent report [4] revealed that over 1.3 billion tons of foods is wasted each year [5]. Despite the reduction in fruit and vegetable waste globally (from 60% in 2011 to 45% in 2015), there is still a need for an improvement in bio-waste utilization [5,6]. Fruit waste and scrapped value are affected by the type of fruit, processing methods, and post-harvest technologies. For example, the discarded portion of banana is 35% [7], pineapple is 33–46% [8], papaya is 15–20% [9], mango is 25–40% [10], citrus is 25–35%, apple is 9–13%, and watermelon is 43–48% [7,11].

Fruits are commonly used fresh and processed into juice, frozen fruit pulp, jam, syrup, and concentrated or dehydrated forms [12]. An enormous amount of waste is generated during fruit processing, and proper disposal is associated with higher operational and transportation costs. Thus, imprudent disposal has negative impacts on both the economy and the environment. Likewise, the wastes generated from fruit processing have environmental risks but also represent an enormous loss of nutrients with high bioactive properties [13]. Fruit by-products including skins, cores, stems, shells, stones, and seeds account for 50–60% of fresh fruit. In most cases, nutritional comparison shows that the by-products, including peel and seeds, have higher nutritional values as compared to the pulp [14]. Additionally, fruit by-products such as skins are natural sources of soluble and insoluble dietary fiber, pectin, and phenolic antioxidants [15,16]. The therapeutic properties of these by-products offer a high potential for further value while assisting in the prevention of non-communicable diseases such as diabetes, cardiovascular disease, obesity, and cancer [17–19]. Through the process of valorization, promising potentials of by-product utilizations in the food, pharmaceutical, biotechnology, and related industries may inspire the food, pharmaceutical, biotechnology, and related industries in the near future. This article reviews fruit by-products and their industrial applications for nutritional and health benefits.

## 2. Sources of Literature

Quantitative and qualitative research findings were used to synthesize the potentials of fruit by-products from the existing literature. This was performed by identifying the published literature indexed in Google Scholar®, Scopus®, Web of Science®, Springer Link, Science Direct, PubMed, and MDPI databases using the following keywords: bioactive components, biotechnological techniques, fruit processing, waste utilization, fruit wastes and by-products, and industrial waste. As some keywords gave a very large number of published articles with a scope far different from the scope of this review, more restricted terms such as "bioactive compounds from fruit by-products," "biotechnological techniques of fruit by-product valorizations," "banana by-product," "avocado by-product," "apple by-product," "citrus by-product," and "watermelon by-product" were used. Articles published between 2000 and 2023 were selected for this review.

## 3. Description of Some Common Fruit By-Products: Overview

### 3.1. Banana (Musa spp.) By-Products

Banana, grown in tropical and subtropical biospheres, is a prominent fruit crop consumed throughout the world. The banana fruit is available year-round at a fair price. Banana processing generates a lot of industrial by-products such as peels, rhizomes, stems, leaves, sheaths, and inflorescence [20]. These by-products are useful resources for various industrial applications, including in the food and medicine manufacturing industries, due to the rich phenolic composition of the peel [21]. Figure 1 shows a banana peel, which is one of the banana by-products.

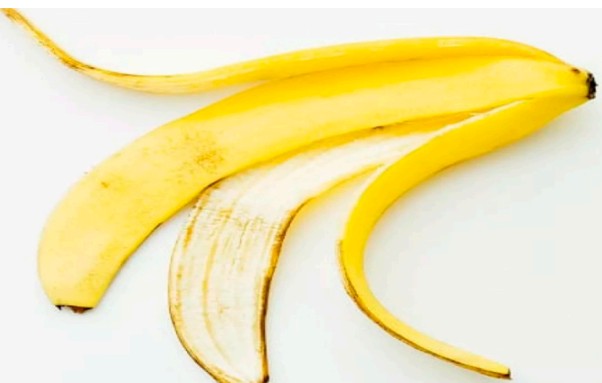

**Figure 1.** Banana peel: Reprinted/adapted with permission from Ref. [22]. 2016, Priwo.

### 3.2. Apple (Malus Domestica Borkh.) By-Products

Apple is among the widely cultivated temperate fruits with pleasing taste, aroma, and health-enhancing substances [23]. Nearly 68% of apples are consumed raw, and the remaining are industrially processed for juice, cider, and powder, generating various by-products such as seed, peel, core, and stem using microwave-assisted phosphoric acid activation [24]. The apple by-products, including seeds and peels, represent roughly 25–30% of the load of the first fresh fruit [25] and are used as ingredients in food formulation [26]. Figure 2 shows an apple by-product.

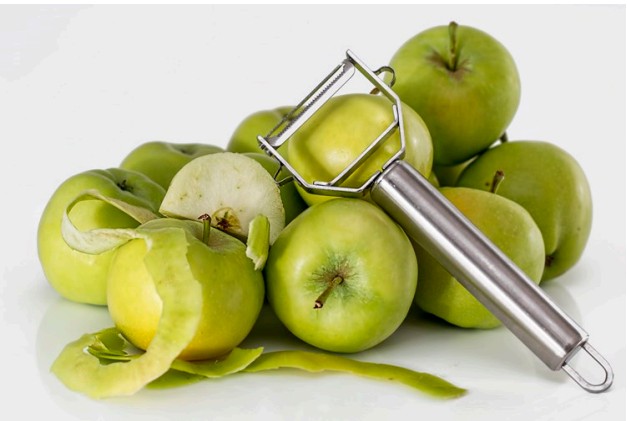

**Figure 2.** Apple by-product: Reprinted/adapted with permission from Ref. [27]. 2022, AITC Canada.

### 3.3. Mango (Mangifera indica L.) By-Products

Mango is a popular tropical fruit with very extensive production. It is called "the king of fruits" because of its luscious flavor, pleasing aroma, and nutritional value [28]. Mango fruits have bio-functional properties and are commonly consumed as fresh, frozen, juice, jam, or nectar [29]. Processing mango fruit into different value-added products creates immense waste products that range from 40 to 60% depending on the variety and the size, of which peels represent 12–16%, seeds 10–25%, and kernels 15–20% [30].

Mango peels are the main by-product generated from industrial processing and fresh consumption and account for 10–20% of the total weight of the mango fruit [31]. Recently, mango peels have gained attention from the scientific community because of their high content of bioactive compounds such as polyphenols, catechins, kaempferol, gallic acid, mangiferin, quercetin, and benzoic acid, with functional and health properties [32,33]. Mango seeds and kernels are additional by-products obtained during the industrial processing of mango. Figure 3 shows mango by-products.

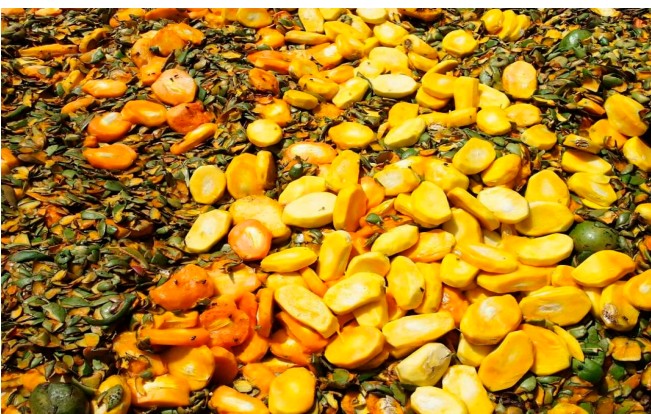

**Figure 3.** Mango by-products: Reprinted/adapted with permission from Ref. [34]. 2023, CIRAD.

*3.4. Citrus (Citrus rutaceae L.) By-Products*

Citrus fruit belongs to the *Rutaceae* family, which include fruits such as oranges, tangerines, mandarins, lemons, limes, sour oranges, and grapefruits. Citrus fruits are among the most widely consumed fruits around the world [35]. Citrus fruits are so-called fleshy fruits with lofty amounts of citric acid, which gives them an acidic taste. Industrially, citrus fruits are processed into juice, jam, marmalade, fruit cocktail, and flavoring agents [36,37].

During processing of citrus fruits, 1–10% of the seed, 60–65% of the peel (flavedo and albedo), and 30–35% of internal tissues (juice sac residues and rag), representing 50–70% of the processed fruit depending upon the variety, processing methods, and growth conditions, were discarded from the total generated by-products [38]. Citrus peel is also dried and mixed with pulp to produce molasses for cattle feed. Figure 4 shows citrus by-products.

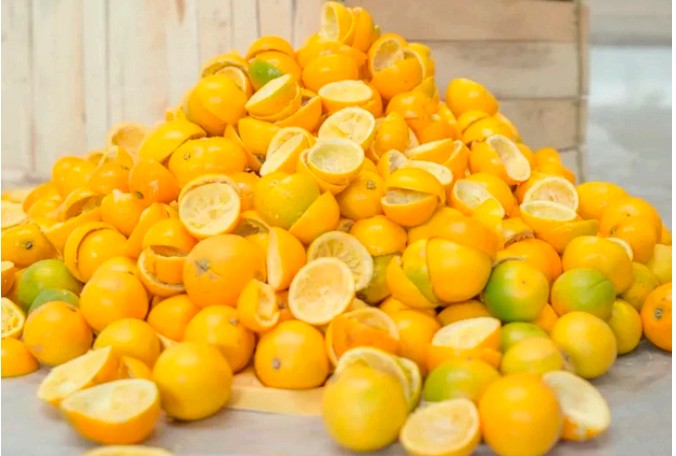

**Figure 4.** Citrus by-product: Reprinted/adapted with permission from Ref. [39]. 2017, BBC.

*3.5. Grape (Vitis vinifera) By-Products*

Grape is a common fruit that is consumed almost everywhere in the world [2]. Approximately 50% of the world's production of grapes goes into winemaking or vinification, and the remaining 50% is consumed fresh or dried to make grape raisins [40]. The main by-products of wine processing are pomace (skins, stems, and residual pulp) and grape seed, which accounts for nearly 20% of the original grape weight [41]. The grape seeds account for about 5% of the total weight of the whole grape; but they account for almost 40–50%, and the pomace accounts for 10–15%, of the discarded solid residues from the wine-processing industries [42]. Figure 5 shows grape by-products.

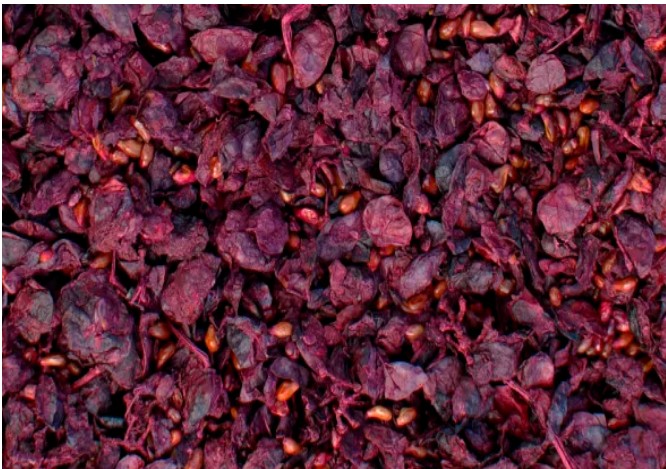

**Figure 5.** Grape by-product: Reprinted/adapted with permission from Ref. [43]. 2017, Chiara Cecchini.

*3.6. Avocado (Persea americana) By-Products*

Avocado is the most important fruit cultivated in tropical and subtropical regions of the world, where one-third of the production is handled by Mexico [44]. Avocado is considered a butter pear due to its shape and the soft texture of its pulp. Avocado fruit contains vitamins (B vitamins, vitamin K, vitamin E, and vitamin C), minerals (potassium, copper), proteins, fibers, phenolic acids, hydroxycinnamic acids, and essential fatty acids (EFA), all of which have substantial health benefits [45]. During industrial processing of avocados into oils, the remaining residues, such as seeds and peels, representing 21–30% of the fruit, are discarded [46]. Figure 6 shows avocado by-products.

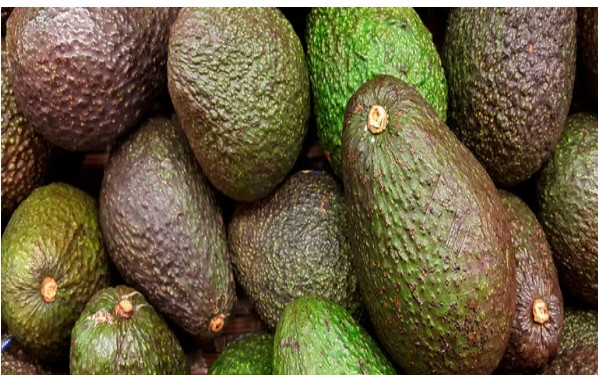

**Figure 6.** Avocado with peel: Reprinted/adapted with permission from Ref. [47]. 2012, Donia Hilal.

*3.7. Pineapple (Ananas comosus L.) By-Products*

Pineapple is a tropical fruit grown in several parts of the globe, with Thailand, Brazil, the Philippines, Costa Rica, and India being the main producers. Asia is the main continent producing pineapples (48.2%), followed by America (34.5%), and Africa (16.4%) [2]. Pineapple is commonly consumed fresh, is used in salads, and is commercially available in juice forms, jams, dehydrated products, and canned foods. Pineapple is added to various types of fruit concentrates because of its neutral color, pleasant flavor, and good acidity/sweetness ratios [48].

Processing pineapple into value-added products generates different by-products, such as residual pulp, peels, stems, and a core, which represent 35–46% of discarded residues [8]. Peels have the highest percentages of by-products, ranging from 30% to 42% $w/w$, followed by the core at around 10% $w/w$. Core and stem share 5% by weight of the total waste. Almost half of the total pineapples produced are discarded, along with plenty of bioactive compounds [48]. Figure 7 shows a pineapple fruit.

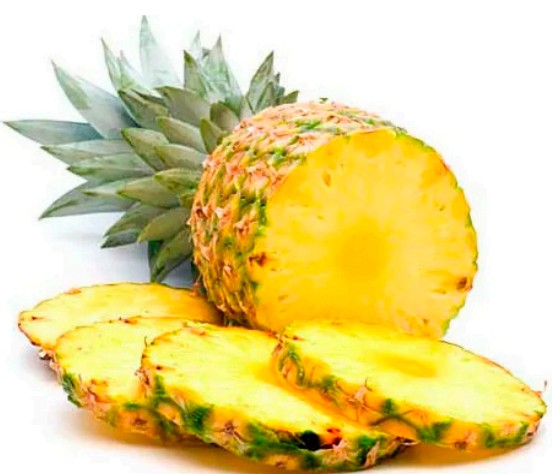

**Figure 7.** Pineapple fruit with skin: Reprinted/adapted with permission from Ref. [49]. 2014, Rico.

## 4. Nutritional Potentials of Fruit By-Products

Fruit by-products, including peel, skin, seeds, pomace, and stones, contain a high amount of bioactive compounds and are good sources of nutrients, including pectin, proteins, fat, fiber, minerals, and vitamins. This section briefly explains the nutritional composition as well as the bioactive compounds found in fruit by-products. Proximate composition, mineral content, vitamin composition, and bioactive compounds extracted from fruit by-products are summarized in Tables 1–4, respectively.

**Table 1.** Proximate composition (%) of some common fruit by-products.

| By-Product | Moisture | Ash | Protein | Fat | Carbohydrate | Fiber (%) | Reference |
|---|---|---|---|---|---|---|---|
| Apple pomace | 8–10.6 | 1–6.1 | 3–5.67 | 1.2–3.9 | 48–62 | 4.7–51.1 | [50] |
| Ripe mango peel flour | 7.86 | 4.5 | 4.1 | 4 | 29.4–32 | 51.1–54.2 | [51] |
| Banana peel | 13.6 | 9.83 | 5.53 | 23.93 | 32.39 | 14.83 | |
| Avocado peel | - | 3 | 4.5 | 4.6 | 72 | 3.8 | [52] |
| Avocado seed | 67.2 | 2.3 | 9.6 | 3.9 | - | 10.7 | |
| Pineapple raw peel | 82.7 | 5.0 | 8.9–9.2 | 1.3 | - | 16.3 | [53] |
| Papaya raw peel | 86.8 | 11.6 | 20.27 | 2.3 | - | 18.5 | |
| Papaya seed | 5.8 | 6 | 23.6 | 23.5 | - | 47.2 | |
| Grape pomace | 3.37 | 4.68 | 8.49 | 8.16 | 29.20 | 46.17 | [54] |
| Citrus peel | 2.49 | 13.20 | 0.42 | 9.74 | 71.57 | 2.58 | [55] |

### 4.1. Banana Fruit

Banana peels represent 30–40% of the edible part of the fruit and are a rich source of phytochemicals such as phenolic compounds, dietary fibers, and carotenoids that are known to have high antioxidant capacity [56]. Banana pulp and peel flour also contain many other phytochemicals, such as catecholamines, flavonoids, phenols, steroids, phytosterols, glycosides, and terpenoids [57]. Additionally, banana peel has macro-minerals (potassium, calcium, phosphorus, and magnesium), trace minerals (iron and zinc), and vitamins (vitamins C and A) [58]. The fiber content of the peel is approximately 50%, and it is a rich source of pectin, which forms gels used as an emulsifier [59]. The nutrient components, including proximate (Table 1), mineral (Table 2), and vitamin (Table 3) contents, are presented and discussed. Bioactive compounds of banana by-products are presented in Table 4, and Table 5 shows their applications in different industries.

### 4.2. Apple Fruit

Apple pomace is a rich source of beneficial bioactive compounds, including phenolic acids, flavonoids, and dihydrochalcones [60]. Apart from its minor components, apple pomace is also a good source of carbohydrates, proteins, vitamins, and minerals (Table 4). The pomace consists mainly of insoluble sugars, including cellulose, hemicellulose, and lignin (a non-starch polysaccharide), with simple sugars such as glucose, fructose, and galactose [61]. It also contains minerals such as phosphorus, calcium, magnesium, and iron. Additionally, apple pomace contains fatty acids, including linoleic acid (18:2 n-6) and oleic acid (18:1 n-9) [50]. The proximate, mineral, and vitamin contents are presented in Tables 1–3, respectively. Bioactive compounds of apple by-products are presented in Table 4, and Table 5 shows their applications in different industries.

**Table 2.** Mineral content (mg/100 g) of some common fruit by-products.

| By-Product | Phosphorus | Potassium | Calcium | Sodium | Magnesium | Iron | Zinc | Copper | Manganese | Reference |
|---|---|---|---|---|---|---|---|---|---|---|
| Apple pomace | 64.9–70.4 | 398.4–880.2 | 55.6–92.7 | 185.3 | 18.5–333.5 | 2.9–3.5 | 1.4 | 0.1 | 0.4–0.8 | [50] |
| Mango seed kernel | 20 | 158 | 10.21 | 2.7 | 22.4 | 1–12 | 1.1–5.6 | 8.6 | 0.04 | [62] |
| Banana peel | - | 4599.7 | 2011.5 | 32.5 | 95.1 | 2.5 | 2.3 | 12.4 | 5.7 | |
| Avocado peel | - | 899.8 | 679.3 | 21.1 | 46.9 | 2.3 | 1.6 | 14.5 | 1.4 | [53] |
| Avocado seed | - | 1202.6 | 434.9 | 39.4 | 55.8 | 3.7 | 1.8 | 16.7 | 1.5 | |
| Citrus peel | 366.84 | 8.75 | 515.78 | 274.77 | 5.39 | 9.06 | - | - | - | [55] |
| Pineapple peel | | 1349.5 | 4236.2 | 9.8 | 107.6 | 1.6 | 0.8 | 4.7c | 8.2 | [53] |
| Grape pomace | 193 | 4334 | 182 | 131 | - | - | - | - | - | [63] |

### 4.3. Mango Fruit

Mango by-products are important sources of bioactive compounds, including vitamin C, beta-carotene, polyphenols, and dietary fiber [64]. Mango seed kernels are used for making multipurpose nutraceuticals owing to their high composition of phytochemicals such as phenolic acids, flavonoids, catechins, hydrolyzable tannins, and xanthanoids [29,30]. Mango seed kernel powder contains a good amount of fat, protein, and carbohydrate, which implies a possibility to produce energy-rich functional foods from this resource [16]. In addition, mango peel is an excellent source of dietary fiber (45 to 78%) and other components such as phenolic acids, flavonoids, xanthones, carotenoids, ascorbic acid, and tocopherols [65]. The proximate, mineral, and vitamin contents are presented in Tables 1–3, respectively. Bioactive compounds of mango by-products are presented in Table 4.

### 4.4. Citrus Fruit

Among the generated wastes of citrus fruit, 60–65% comes from the peel, 30–35% from interior tissues, and 0–10% from seeds [66], which provide a valuable source of phytochemicals with high antioxidant activity, anti-inflammatory properties, and anticancer properties compared to the edible portion [67]. The proximate, mineral, and vitamin contents are presented in Tables 1–3, respectively. Table 4 lists the major bioactive compounds found in citrus by-products.

### 4.5. Grapes

Wine-processing by-products have various types of biomolecules (dietary fibers, lipids, proteins, and natural antioxidants and phenolic compounds) and are a cheap source for the development of dietary supplements [68]. Grape seed is one of the major by-products, with dense bioactive components, including stilbene, resveratrol, gallic acid, rutin, and catechinalate. These bioactive compounds exhibit cardiovascular-protective, antimicrobial, antioxidant, and anticancer properties [69].

## 4.6. Avocado Fruits

Fresh avocado peel is a potential source of carbohydrates (60–73%), proteins (3–8%), lipids (4–9%), fiber (about 50%), and ash (3–6%). Similar to the peel, the avocado seed also consists of 72% carbohydrates, 4.5% proteins, 4.6% lipids, 3.8% fiber, and 3% ash [52]. Avocado peel and seed are also high in phytochemicals such as phenolic acids, condensed tannins, and flavonoids [70]. These bioactive compounds have been shown to exhibit antioxidant and anti-inflammatory properties. Furthermore, avocado by-products have numerous applications in different industries, as shown in Tables 5–7.

## 4.7. Pineapple

Pineapple by-products are low-cost sources of dietary fiber, which may be applied to the production of fiber-rich foods [71]. Pineapple by-products are also used as a substrate for the production of organic acids, with great commercial demand for acidification and flavor-enhancing of low-acid foods, including most fresh vegetables and fruits [72]. The proximate, mineral, and vitamin content are presented in Tables 1–3, respectively. Table 4 shows the bioactive compounds of pineapple by-products, and their utilization in different industries is summarized in Tables 5–7.

**Table 3.** Vitamin content of some common fruit by-products in (mg/100 g).

| Vitamin (mg/100g) | Fruit By-Products | | | | | |
|---|---|---|---|---|---|---|
| | **Citrus Peel** | **Mango Seed** | **Avocado Seed** | **Pineapple Stem** | **Grape Pomace** | **Pineapple Peel** |
| Vitamin B1 | 11.9 | 0.08 | 0.33 | - | - | - |
| Vitamin B2 | - | 0.03 | 0.29 | - | - | - |
| Vitamin B3 | 234.16 | - | 0.06 | - | - | - |
| Vitamin B6 | 286.63 | 0.19 | - | - | - | |
| Vitamin B9 | 1.36 | - | - | - | - | - |
| Vitamin B12 | - | 0.12 | - | - | - | - |
| Vitamin C | 21.34 | 0.56 | 97.8 | 121.2 | 26.25 | 212.9 |
| Vitamin A | - | 15.27(IU) | 10.11(IU) | - | - | - |
| Vitamin E | 4.45 | 1.3 | 0.12 | - | - | - |
| Vitamin K | - | 0.59 | - | - | - | - |
| Reference | [55] | [73] | [74] | [75] | [54] | [75] |

## 5. Health Benefits of Fruit By-Products

Fruit by-products contain phytochemicals such as phenolic compounds, vitamins, minerals, dietary fiber, and other bioactive compounds. The polyphenolic compounds found in fruit promote human health development. The phenolic compounds are the secondary metabolites of fruits that can act against free radicals and oxidative stresses, and thus they are known as antioxidants [76,77].

### 5.1. Health Benefits of Banana By-Products

Banana peel, the fruit's primary by-product, accounts for approximately 35–40% of the fruit's total mass [78,79]. Banana peel contains high amounts of dietary fiber and phenolic compounds, as well as antioxidant, antibacterial, and antibiotic properties. As a result, it is a material with significant potential, which encourages its application in the nutraceutical and pharmaceutical industries [78]. In terms of nutritional quality, banana peel has shown excellent uses in a variety of food items such as bakery, culinary products, and meat products, owing to the presence of various bioactive compounds that may have health-promoting properties [80]. Furthermore, banana peels are high in polyunsaturated fatty acids such as linoleic acid (Omega-6) and α-linolenic acid (Omega-3), which account for more than 40% of total fatty acid content [81]. Linoleic acid has been shown to reduce liver fat and slightly improve metabolic status without causing inflammation. In addition, clinical research has shown that α-linolenic acid has an anti-inflammatory effect on obesity [80].

Bioactive compounds found in banana peels include flavonoids, tannins, phlobatannins, alkaloids, glycosides, anthocyanins, and terpenoids, which have antibacterial, antihypertensive, antidiabetic, and anti-inflammatory properties [82]. Phenolics are important secondary metabolites found in higher concentrations in banana peels compared to other fruits. Banana peel contains a variety of phenolic compounds, including gallic acid, catechin, epicatechin, tannins, and anthocyanins [79]. Furthermore, gallocatechin levels in banana peel are five times higher than in pulp, indicating that the peel is a rich source of antioxidant compounds. Flavonols, hydroxycinnamic acids, flavan-3-ols, and catecholamines are the four subgroups of phenolic compounds found in banana peel. The proposed mechanism of these phenolic compounds' antioxidant effect involves preventing reactive oxygen species (ROS) formation, direct ROS scavenging, and induction of antioxidant enzymes [83].

Several studies have linked ROS to a variety of chronic diseases, including neurodegeneration, cancer, diabetes, and inflammation [84]. Rutin and myricetin are the most abundant phenolic compounds in plantain and dessert banana peel flavonol profiles [85]. According to Phacharapiyangkul et al. [86], ferulic acid, which is abundant in sucrier banana peel, may act as an anti-melanogenesis factor by regulating vascular endothelial growth factor expression, initiating nitric oxide synthase, and acting as a tumor suppressor gene.

### 5.2. Health Benefits of Apple By-Products

Apple pomace is one of the most commonly produced agri-food wastes; however, the pomace produced by apple processing can be reused in biotechnological processes as a substrate for the production of various compounds such as flavoring compounds, pigments, fuel, and citric acid, or as raw material for fiber and phenolic compound extraction [87]. From a nutritional standpoint, apple pomace is a by-product high in fibers, vitamins, minerals, phenolic compounds, and pigments. All of these macronutrients play an important role in the human body due to their effects on metabolism [88]. These components can help to treat gastrointestinal disorders, lower serum triglycerides and LDL cholesterol, and regulate glycemia. All of these effects on the human body can be explained by their high concentration of the beneficial compounds mentioned above, which primarily play anti-inflammatory and antioxidant roles [89].

Dihydrochalcones, procyanidins, flavan-3-ol monomers, flavonols, anthocyanidins, and hydroxycinnamic acids are the most abundant phenolic compound families in apple pomace. Phlorizin from the dihydrochalcones family, chlorogenic acid from the hydroxycinnamic acids family, and epicatechin from the flavan-3-ol monomer family are the most representative compounds [88]. Phlorizin is a remarkable phenolic compound found in apple pomace that acts as a strong antioxidant, anti-inflammatory, and antimicrobial agent. Furthermore, phlorizin has several health benefits, most notably in diabetes, due to its ability to change the way glucose is absorbed and excreted. Furthermore, research shows that phlorizin specifically and completely inhibits sodium/glucose cotransporters in the intestine and kidneys. This property may benefit postprandial hyperglycemia therapy in diabetes and other related illnesses such as obesity [67,88].

### 5.3. Health Benefits of Mango By-Products

Mango by-products are excellent sources of phytochemicals with broad bioactivities that ultimately improve consumers' health. Mango peel accounts for 7–24% of the total weight of a mango fruit. Mango peel has piqued the scientific community's interest due to its high content of valuable compounds such as phytochemicals, polyphenols, carotenoids, enzymes, vitamin E, and vitamin C, all of which have functional and antioxidant properties. These valuable compounds are also advantageous to human health. Many researchers have reported that mango peels can be used to produce valuable ingredients (such as dietary fiber and polyphenols) for a variety of food applications [90]. Mango peels have yielded two major valuable compounds: ethyl gallate and penta-O-galloyl-glucoside. These compounds have strong scavenging activities for hydroxyl radicals (-OH), superoxide anion ($O_2^-$), and singlet oxygen ($1O_2$). Mango peel waste has the potential to be used in both

experimental and clinical settings. Meanwhile, pharmaceutical studies have shown that gallate-type compounds, such as penta-O-galloyl-glucoside, have anti-tumor, antioxidant, anti-cardiovascular, and hepatoprotective properties [91].

### 5.4. Health Benefits of Citrus By-Products

Citrus by-products have high moisture content and high organic matter content. Furthermore, they are high in sugars (glucose, fructose, sucrose), carbohydrates (cellulose, starch, pectin, dietary fibers), proteins, organic acids (citric, malic, oxalic acids), lipids (linolenic, oleic, palmitic, stearic acids), essential oils (limonene), pigments/carotenoids (carotene, lutein), and vitamins (flavonoids, phenolic acids) [92]. Furthermore, citrus by-products are high in biologically active compounds such as polyphenols, flavonoids, and phenolic acids. It is worth noting that citrus by-products contain more polyphenols than the edible portion of the fruit. As a result, in recent years, the extraction of polyphenols from citrus by-products has piqued the interest of many researchers due to their massive quantities and multifaceted properties such as antioxidant, anti-inflammatory, and anticancer effects, among others [93,94].

Flavonoids are the most diverse class of polyphenols found in citrus by-products, offering a wide range of health benefits as well as excellent antioxidant properties. Citrus by-products are high in phenolic acids, in addition to flavonoids. These compounds are classified into two subgroups based on their free radical scavenging activity: hydroxybenzoic (gallic, vanillic, and syringic acids) and hydroxycinnamic acids. (caffeic, ferulic, p-coumaric, and sinapic acids). According to various studies, the total flavonoid and total phenolic content of citrus fruits can vary depending on the species, cultivars, and harvesting conditions [95,96].

### 5.5. Health Benefits of Grape By-Products

Grape pomace contains nutrients such as carbohydrates, fibers, minerals, and vitamins. Dietary fiber is found in high concentrations among these nutrients. Several studies have shown that grape pomace contains up to 70% total dietary fiber, with insoluble dietary fibers such as cellulose and hemicellulose accounting for the remaining 26 to 78%. Water-soluble dietary fiber (DF), which includes α-glucans, pectins, gums, and so on, accounts for approximately 9–11% of pomace [97]. Pomace fiber's physiological health benefits are related to its ability to ferment in the colon, producing short chain fatty acids that act as a prebiotic. Aside from these nutrients, grape pomace contains a variety of non-nutrient components known as bioactive compounds, the most important of which are phenolic compounds [98].

Grape pomace is a rich source of phenolic compounds such as monomeric phenolic acids, oligomeric proanthocyanidins, and glycosylated anthocyanins, all of which have antioxidant and antimicrobial properties [99]. They are also a potential source of catechins, epicatechin, dimers and trimers of procyanidins, and resveratrol. Anthocyanins are pigments found in grape pomace [98]. Anthocyanins are antioxidants and antimutagenics. Stems contain a high concentration of tannic compounds with nutraceutical and pharmacological potential. The most common phenolic compounds found in grape pomace are hydrobenzoic and hydrocynnamic acids, flavonols, stilbenes, and anthocyanins [100]. In general, GP extract has been extensively studied for its wide range of activities, including cardio-protective, anticancer, anti-inflammatory, anti-aging, antimicrobial, and other health-promoting properties [101]. Furthermore, grapes and their by-products are high in dietary fiber, which has been linked to a variety of health benefits such as glucose absorption regulation, obesity prevention, blood cholesterol reduction, and reduced cardiovascular risk. In addition, grape pomace is a good source of fiber for the industry, with a higher potential for regulating bowel functions and water retention [102,103].

*5.6. Health Benefits of Avocado By-Products*

Avocado by-products are high in carbohydrates, lipids, proteins, dietary fiber, vitamins, minerals, and phenolic compounds [104]. The composition of avocado by-products clearly shows that these biomasses have enormous potential as a source of valuable compounds with applications in a variety of industrial sectors. Bioactive compounds have been identified in particular, including phenolic compounds (hydroxycinnamic acids, hydroxybenzoic acids, flavonoids, and proanthocyanidins), carotenoids, alkaloids, acetogenins, and phytosterols [104,105]. These compounds can be used as nutraceuticals, but they also have applications in the food, health, pigment, and material industries [105].

Avocado seeds (an industrial by-product) have anti-inflammatory, antioxidant, and antimicrobial properties that can be used to prevent and treat gastric disorders [106]. Anticholinesterase and antioxidant activities were found in *P. americana* leaves and seeds [107]. The phenolic components and antioxidant activity of avocado skin and seed hydroethanolic extracts revealed a predominance of compounds from the flavonoid, proanthocyanidin, and hydrocinnamic acid groups. Avocado pulp contains phenolic compounds such as gallic acid, 3,4-dihydroxyphenylacetic acid, 4-hydroxybenzoic acid, vanillic acid, p-coumaric acid, ferulic acid, and quercetin, which have antioxidant properties [108]. Avocado peels from var. Colinred showed the highest total phenolic content, and specifically B-type procyanidins and epicatechin, as well antioxidant activity, when compared with seeds. These authors also showed that the peel extract can protect the transgenic parkin Drosophila melanogaster fly against paraquat-induced oxidative stress, movement impairment, and lipid peroxidation, as a model of Parkinson's disease [109]. Antiradical activity of avocado by-product is mainly due to polyphenols (+)-catechin, (−)-epicatechin, 3-O-caffeoylquinic acid (chlorogenic acid isomer), and three compounds of the flavonoid family. Flavonoids were the most abundant group in avocado seed and seed coat, with quercetin, ()-naringenin, and sakuranetin being the most abundant. Phenolics and flavonoids are bioactive compounds that have been linked to a reduction in a variety of deteriorative processes in the human body due to their ability to reduce free radical formation and to scavenge [110].

*5.7. Health Benefits of Pineapple By-Products*

Pineapple peel accounts for about 57% of the total by-product and contains a high concentration of insoluble dietary fiber, vitamins, minerals, phenolic compounds, and other bioactive compounds with high antioxidant capacity [111,112]. Because of their biological properties with applications in human health, phenolic compounds derived from pineapple by-products are of great interest in the pharmaceutical and food industries. There have been few studies on bioactive polyphenols derived from pineapple residues. Bioactive polyphenols of pineapple by-products include myricetin, salicylic acid, tannic acid, trans-cinnamic acid, and p-coumaric acid, which were discovered in a high dietary fiber powder made from pineapple shell, which is a by-product, and these compounds have been reported to be potent antioxidants [72,113]. Polyphenols found in pineapple wastes, such as ferulic acid and syringic acid, have been shown to have antioxidant and antimicrobial activity [72].

## 6. Extraction Methods of Fruit By-Products

The most widely used technique at the industrial scale is conventional solvent extraction, which includes several phases such as solid–liquid extraction (e.g., Soxhlet) using organic solvents, maceration, and hydrodistillation [77,114]. These methods, however, have the potential to degrade thermolabile compounds. To address this issue, the food industry is interested in extraction techniques such as enzyme-assisted, ultrasound-assisted, microwave-assisted, pulsed-electric-field assisted, pressurized liquid extraction, and supercritical fluid [80]. Furthermore, various innovative technologies are now being used for extracting valuable compounds from fruit waste and by-products [77,115].

### 6.1. Enzyme-Assisted Extraction

Cellulase, α-amylase, α-glucosidase, xylanase, α-glucanase, pectinase, and other related enzymes are used to improve the extraction process by hydrolyzing the matrix of the plant cell wall, resulting primarily from the formation of the enzyme–substrate complex, during which bonds in the substrate molecules break into the final products [116]. The sizes of the plant material, enzyme concentration, reaction time, temperature, pH, and solid–liquid ratio all have an effect on the enzyme–substrate complex [117]. Enzyme-assisted extraction of lycopene from industrial tomato waste has been reported [118].

### 6.2. High Hydrostatic Pressure (HHP)

HHP is a food-processing technology that generates and sustains high pressures using special equipment (100–1000 MPa). A high-pressure vessel, its head closure, a pressure generation system, and a temperature control device comprise a typical HHP system [119,120]. The pressure vessel is the heart of the HHP system, and the thickness of its walls determines the maximum working pressure. The maximum working pressure varies from 400 to 600 MPa depending on the internal diameter of the vessel. Pre-stressed vessel designs, such as multilayer vessels or wire-wound vessels, are used in cases of higher pressures [120].

HHP is widely applied for microbial inactivation and food preservation. However, more recently, the technology's potential for extracting valuable compounds from food waste has been confirmed [121,122]. The damage to the fruit cellular structure caused by high hydrostatic pressure can improve the mass transfer rate, increase solvent permeability, decrease processing time, and, as a result, achieve high extraction yields [123,124]. HHP is successfully used to extract valuable bioactive components from different fruits including papaya seed [125], Cape gooseberry pulp [126], and grape skin pomace [127].

### 6.3. Membrane Separation

Membrane technology operates on a thin physical barrier through which materials can pass (the permeate) or be retained (the retentate) in response to a driving force that can be a difference in pressure, concentration, temperature, and/or electrical potential. A membrane's separation performance is influenced by its inherent properties, such as its chemical composition, as well as process variables such as temperature, pressure, and feed flow. Furthermore, interactions between feed-flow components and the membrane surface must be considered [120,128,129]. The main physical operational parameters influencing the permeate flow rate are pressure, temperature, viscosity, density of the feed fluid, and tangential velocity. Membrane technology has been successfully used to purify and concentrate bioactive compounds from fruit-processing wastes, with potential applications in food colorants, food supplements, pharmaceutical applications, and cosmetic products [77,129,130].

### 6.4. Microwave-Assisted Extraction

Microwaves have frequencies ranging from 300 MHz to 300 GHz; in microwave-assisted extraction, the small amount of moisture present in a plant cell is heated, causing evaporation and creating enormous pressure on the cell wall, further weakening and breaking it and allowing the phytoconstituents to be released to the outside [131]. It is a relatively new application that uses microwave energy to extract soluble solids from a variety of materials. It has been designated as a green technology because it reduces the use of organic solvents [123,132]. Microwave-assisted extraction has grown in popularity for recovering low molecular weight organic compounds or small molecules from food matrices [120,133]. Several factors influence microwave-assisted extraction of natural bioactive compounds from fruit wastes, including power, frequency, processing time, sample moisture content and particle size, extraction temperature, pressure, extraction cycles, type of solvent, and solid sample to liquid solvent ratio [80]. The technology is widely applied to extract various bioactive components from different fruit by-products, such as pomegranate fruit peel [134], black currant by-product [135], mango peel [136], and so on.

### 6.5. Pressurized Solvent Extraction

Pressurized solvent extraction uses solvents at high pressures and temperatures above their boiling points. High temperatures (between 100 and 374 °C) positively impact the mass transfer rate, surface equilibria, and extraction rate, while high pressure (typically ranging from 4 to 20 MPa) prevents solvent evaporation and also influences the mass transfer of the solvent into the pores matrices and, thus, analyte solubility [120,132]. Accelerated solvent extraction and subcritical solvent extraction are also terms for this technology. While all of the solvent is water, the technique is known as superheated water extraction, subcritical water extraction, or pressurized hot water extraction. Water is the most commonly used solvent in this technology because water has a high diffusivity, low viscosity, and low surface tension under subcritical conditions. This improves the kinetics of mass transfer and the solute's solubility [120,137,138]. For the first time, Wijngaard and Brunton [139] successfully applied the technology to extract bioactive compounds (antioxidants and polyphenols) from apple pomace.

### 6.6. Pulsed Electric Fields

The use of pulsed electric fields (PEF) as an extraction technique for the purpose of recovering bioactive compounds has received little attention thus far. The technology involves applying an external electrical field to food placed between two electrodes for a few microseconds. When a biological cell (plant, animal, or microbial) is exposed to high intensity fields (kV/cm) in the form of very short pulses (ms to ms), temporary or permanent pores form on the cell [120,140]. This phenomenon, known as electroporation, causes cell membrane permeabilization, or an increase in permeability, and if the intensity of the treatment is high enough, cell membrane disintegration occurs. The degree of permeabilization achieved, and thus the treatment intensity, is affected by a number of process parameters, including the electric field strength and the number, duration, and shape of pulses [141]. This type of process could aid in the development of quality-preserving preservation and extraction processes in the food industry [115,120]. The electric field intensity, pulsed wave shape, solvent selection, raw material to solvent ratio, pulse duration, and treatment temperature all have a direct impact on the effectiveness of PEF treatment [115]. Pulsed electric field is used to extract bioactive compounds in blueberry by-product [142], thinned peach by-product [143], and plum and grape peel [144].

### 6.7. Supercritical Fluid Extraction

Supercritical fluid extraction (SFE) operates on the use of a fluid at pressures and temperatures above its critical point in order to achieve significant physical changes that alter its solvent capabilities [77]. Although the first experimental works dealing with supercritical fluid extraction date back to the nineteenth century, interest in this technique as an alternative to conventional solvent-based extraction techniques has only recently increased. Carbon dioxide ($CO_2$) has been the most widely used solvent for SFE because of its versatility [145]. $CO_2$ properties can be tuned in supercritical phase (temperature of 31.1 °C and pressure of 7.4 MPa) to provide extracts with desirable compositions. Simultaneously, it ensures a safe separation process for both human health and the environment, with no degradation of heat-sensitive compounds and no toxic solvent residue in the solutes after the process [120,146]. Aside from its physical properties, $CO_2$ is safe, food grade, and widely available at a low cost and high purity [120].

SFE, on the other hand, has some limitations that limit its use. Because it is primarily used to extract non-polar substances, it has a limited capacity for recovering compounds from water-rich by-products. To overcome this limitation, a co-solvent, such as water or ethanol, is usually added. The co-solvent has the effect of increasing the number of polar compounds and intermediate polarity that can be extracted [120]. The primary factors influencing this process are the type of solvent (mostly $CO_2$), temperature, pressure, flow rate, time, and co-solvent concentration (ethanol/water) [115].

Sánchez-Camargo et al. [147] extracted carotenoids from mango peel effectively by using supercritical fluid extraction techniques.

### 6.8. Ultrasound

Ultrasound has been identified as a promising emerging technology that can be used successfully in the extraction field due to its ability to accelerate heat and mass transfer. The process's efficiency is linked to a phenomenon known as acoustic cavitation. Sound waves (frequencies greater than 20 kHz) travel through matter, causing expansion and compression cycles [120,148]. The compression pushes molecules together, while the expansion pulls them apart. Bubbles are formed in the liquid when the ultrasound waves reach a sufficient intensity. Bubbles, once formed, can absorb energy from sound waves, grow during expansion cycles, and recompress during compression cycles. When they can no longer absorb this energy, they collapse, causing shock waves of extreme pressure and temperature (around 100 MPa and 5000 K, respectively) [120].

The collapse of cavitation bubbles near a solid boundary produces high-speed jets of liquid that can strike the food matrix's surface. The cavitation phenomenon causes the solvent to penetrate deeper into the cellular material, improving mass transfer, disrupting biological cell walls, and facilitating compound release [148,149]. Although ultrasound frequency has a significant effect on extraction yield and kinetics, the presence of a dispersed phase can attenuate the ultrasound waves due to differences in compressibility, heat capacity, and thermal diffusion between the droplets of the dispersed phase and the continuous primary phase [115,120,150]. Power, frequency and amplitude, pH, extraction time, extraction temperature, liquid–solid ratio, and particle size are the most important physical parameters in the ultrasound process. Ultrasound has been used to extract bioactive compounds from various plant by-products such as tomato by-product [151].

### 7. Current Knowledge on the Utilization of Fruit By-Products

#### 7.1. Banana Fruit

Banana peel is used to produce livestock feed, fertilizer, biogas, and oil [20]. In addition, it is used as a means of heavy metal removal in water purification [152]. Uses of banana by-products in various industries are presented in Tables 5–7.

#### 7.2. Apple Fruit

About 20% of apple pomace is used traditionally for compost and animal feed, while a large proportion (almost 80%) of it remains underutilized and discarded with a great negative impact on the environment [61].

#### 7.3. Mango Fruit

Mango seed kernels are used for making multipurpose nutraceuticals owing to their high composition of phytochemicals such as phenolic acids, flavonoids, catechins, hydrolyzable tannins, and xanthanoids [29,30]. Mango peel is an excellent source of pectin, enzymes, and fiber for functional food development, while only a few studies show utilization of mango by-products for non-food applications as biosorbents [153]. Biosorbents are biological materials containing a variety of functional sites that have the ability to remove heavy metals such as cadmium (Cd) and lead (Pb) from aqueous solutions [154]. In addition, mango seeds are used as biosorbents to remove heavy metals such as chromium (Cr) [155] and the dye malachite green, which is widely used for food coloring, textile, paper, and acrylic industries [156].

**Table 4.** The main bioactive compounds found in some common fruit by-products.

| Bioactives | Sources | Bioactivity/Preservative | Reference |
|---|---|---|---|
| Flavonols | Pomegranate peels, orange peels, tamarind seeds, mango peels | Antioxidants | [157] |
| Pectins | Kiwifruit, pomegranate, apple, and orange peels | Food additive, thickening agent | [158] |
| Amino acids and proteins | Mandarin by-products, pineapple peels, papaya peels | Good source of protein | [159] |
| Polyphenols | Avocado seed | Antioxidant activity | [44] |
| Phenolic compounds | Banana peel | Antioxidant activity | [21] |
| Triterpenoids | Apple pomace | Anti-inflammatory, antimicrobial, | [25,160] |
| Limonoids | Citrus seed | Anti-inflammatory, anticancer, antibacterial, antioxidant activities | [161] |
| Phenolic acids, flavones, flavanones | Citrus peel and pulp | Antioxidant, anti-inflammatory, anticancer properties | [162] |
| Carbohydrates (pectin and pectin oligosaccharides) | Apple pomace | Dietary fiber, prebiotic, hypo-cholesterolemic | [163] |

*7.4. Citrus Fruit*

Citrus peels are dried and mixed with pulp to produce molasses for cattle feed. Pectin extracted from citrus peels has immuno-modulatory effects on the levels of cytokine secretion in the spleen of mice with a pro-inflammatory potential, as previously reported [164]. Recently, citrus by-products are gaining attention and being valorized by using anaerobic digestion for the production of biogas and fermentation to produce high-value-added chemicals and bio-fuels [165].

*7.5. Grapes*

Most often, grape by-products are used for distillate preparation, animal feed, and compost [40]. Grape seed is one of the major by-products with dense bioactive components. Different uses of grape by-products in various industries are listed in Tables 5–7.

*7.6. Avocado Fruit*

Avocado by-products have many interesting properties that widen their application prospects. Avocado by-products can be utilized for energy production; its pulp oil is used for biodiesel, and its seed oil is used for biodiesel, charcoal, liquid fuels, and fuel additives [52]. Additional utilization of avocado by-products in some industries is listed in Tables 5–7.

*7.7. Pineapple Fruit*

Pineapple processing generates a huge amount of by-products such as residual pulp, peels, stems, cores, and leaves [166], which represent 45–65% of the residuals, which, in most cases, are discarded as waste with significant environmental pollution potential if not properly and efficiently utilized [167]. Pineapple peel is used as a source for the extraction of antioxidant compounds (phenolic compounds such as ferulic acid and vitamins A and C). Successful extraction and recovery of these antioxidant compounds could have implications for the production of antioxidant-rich functional foods [168]. Pineapple by-products can also be used for bioethanol production and bromelain extraction [169].

**Table 5.** Utilization of some common fruit by-products in food industries.

| Fruit By-Product | Uses in Food Industries | Reference |
|---|---|---|
| Apple pomace | Apple pomace used as a dietary fiber source in some baked foods, chicken-meat-based sausages, and yoghurt products | [170,171] |
| Apple pomace | Used as stabilizers for oil–water emulsions and has an antimicrobial activity | [172] |
| Apple seed | Addition of defatted apple seed powder into chewing gum enhanced phloridzin uptake | [173] |
| Avocado by-product | Avocado by-products can be used as antioxidants, antimicrobials, and food additives such as colorants, flavorings, and thickening agents | [14] |
| Avocado peel | Dried peels used in a functional beverage formulation (tea rich in antioxidants) | [174] |
| Avocado peel | Peel extracts used to inhibit lipid peroxidation and to avoid oxidation of meat proteins | [175] |
| Avocado seed | Seed starch used for biodegradable polymers for drug delivery or food pack by-product | [176] |
| | Seeds can act as functional ingredients in foods, considering their composition in total fiber (lignin, cellulose, and hemicelluloses) | [177] |
| Banana peel | The flour obtained from unripe banana peels used for colon health effects due to its high resistant starch content and ripe peels is digestible due to the high content of starch and proteins | [57] |
| Banana peel | Banana peel jelly has antioxidant properties | [178] |
| Citrus peel | Citrus peels used as a source of molasses, pectin, oil, and limone | [179] |
| Citrus (pectin) | Citrus pectin is used as a thickener, emulsifier, and stabilizer in many foods (jams, jellies, marmalades, and other products) | |
| Citrus (pectin) | Pectin is a suitable polymeric matrix for edible films for active food pack by-product | [180] |
| Citrus essential oils | Citrus essential oils are GRAS and are used as antimicrobials, antifungals, and flavoring agents | [181] |
| Grape pomace | Meat and fish derivatives containing grape pomace powders show improved sensory and physical properties | [182] |
| Grape (stems, seeds, and skins) | Fiber from grape pomace used as functional ingredient in bakery products | [183] |
| Grape seed | Oil obtained from grape seed is rich in linoleic acid (60–70%), as well as in tocopherols, which hinder their oxidation | [184] |
| Mango peel | Mango peel powder used as source of antioxidant and dietary fiber in macaroni | [185] |
| Mango peel seed kernel | Mango peels and seed kernel powders used as sources of phytochemicals in biscuits | [186] |
| Mango peel extract | Peel extracts used in gelatin-based films for active food pack by-product due to their free radical scavenging activity and improvements in film strength | [187] |
| Mango peel | Edible films made of mango peel showed good permeability and hydrophobicity properties | [188] |
| Pineapple peel | Pineapple peel is a rich source of sugar that can be used as a nutrient in fermentation processes | [189] |
| Pineapple core | Core can be used in pineapple juice concentrates, vinegar, and wine production | [189] |
| Pineapple stem | Bromelain enzyme extracted from the pineapple stem used as a meat tenderizer, bread dough improver, fruit anti-browning agent, and beer clarifier | [190] |

## 8. Prospective Impact of Fruit By-Products on Food and Nutrition Security

According to the FAO, more than 820 million people in the world are still suffering from hunger in 2018, which underscores the immense challenge of achieving the Zero Hunger target by 2030 [191]. Therefore, the search for alternative food sources for human consumption with high nutritive value is needed. These alternative and innovative food sources would fulfill the need to feed the exponentially growing human population as 70% more food is needed to cover the gap, which becomes an imperative. On one hand, exploring the unexplored, refining the unrefined traits, cultivating the uncultivated, and popularizing the unpopular remain the most appropriate steps proposed by some researchers to achieve food and nutrition security with consideration to the current global food challenges [192–194]. However, a significant amount of by-products from the fruit-processing industry are discarded due to ineffective management and disposal systems. Such fruit by-products have been proven to be rich sources of nutritious and bioactive components and have a considerable effect on the economy and environmental safety. As a result, a careful investigation of the adequate supply of nutritious components from by-products may be of interest and appear to have a positive impact on global food and nutrition security.

**Table 6.** Utilization of common fruit by-products in medicinal and pharmaceutical industries.

| Fruit By-Product | Uses in Pharmaceutical Industries | Reference |
|---|---|---|
| Guava leaf | Guava leaves contain high levels of antioxidants, phenolic compounds, and immune-stimulatory agents | [195] |
| Apple phloridzin | It can inhibit lipid peroxidation and prevent bone loss, enhance memory, and even inhibit cancer cell growth | [196–198] |
| Apple peel | Apple peel consumption improves metabolic alterations associated with a fat-rich diet and also slows atherogenesis development | [199] |
| Avocado peel extracts | Avocado peel extract has been proved to be useful as an inhibitos for the inflammation mediator nitric oxide by a possible reduction of free radicals during inflammation | [70] |
| Avocado peel and seed | Polyphenols from avocado peel and seed possess anticancer, antidiabetic, and antihypertensive effects | [44] |
| Banana peel | The bioactive compounds extracted from peel demonstrate antioxidant, antibacterial, antifungal activity, reduce blood sugar, lower cholesterol, and show anti-angiogenic activity and neuro-protective effect | [21,186] |
| Banana peel | Banana peels are used to synthesize bio-inspired silver nanoparticles, which are used as antimicrobials to pathogenic fungi and some bacterial species | [152] |
| Citrus pulp and seed | D-limonene was shown to exhibit a therapeutic effect on lung cancer in mice and breast cancer in mice and rats | [200,201] |
| Grape by-product | Grape by-products used in pharmaceuticals due to their antibacterial, antiviral, and antifungal properties, they also showed anti-inflammatory actions | [69,202] |
| Grape seed oil | Grape seed oil evaluated in various in vitro and in vivo tests showed antimicrobial, anti-inflammatory, cardio-protective, and anticancer properties | [203] |
| Mango seed and peel | Seed and peel extracts were shown to have anti-inflammatory and antioxidative properties during in vivo studies related to obesity, diabetes, CVD, and skin cancer | [204] |
| Mango pectin | Pectin extracted from mango by-products used for prevention and reduction of carcinogenesis | [205] |
| Mango seed/peel | Mangiferin extracted and isolated from the seed/peel shows strong antioxidant, anti-tumor, antibacterial, and immuno-modulatory effects | [206] |
| Peach kernel | Peach kernel phenols, carotenoids, and cyanogenic glycosides have antidiabetic, antioxidative, and anti-aging properties | [207] |

Proper management of these by-products is believed to be the key opportunity to increase their utilization. This includes cost-effective extraction techniques that give optimum yields of the by-products for their reuse in a wide array of industrial applications. Therefore, it is important to carry out such studies within the realm of fruit regarding the by-products' extraction and wider utilization. This can significantly help to reduce food loss and waste, which can improve food security and environmental sustainability. It has been reported that fruit by-products such as skins, cores, stems, shells, stones, and seeds account for 50–60% of fresh fruit. In most cases, by-products appear to have higher nutritional values than the pulp [14]. Paradoxically, human feeding habits have given preference and priority to a smaller portion of the fruit, resulting in food and nutritional insecurity.

**Table 7.** Utilization of some common fruit by-products in biotechnology.

| Fruit By-Products | Uses in Biotechnology | Reference |
|---|---|---|
| Apple pomace | Apple pomace used as a substrate for value-added products, such as enzymes, aroma compounds, and organic acids | [208] |
| Avocado peel | Carbonaceous material produced from avocado peel is used as alternative adsorbent for dyes removal | [209] |
| Banana peel | Banana peels can be used as substrates by solid state fermentation (SSF) to produce enzymes and organic acids | [210] |
| Banana peel | Organic acids (citric, lactic, and acetic acid) were successfully produced from banana peels with *Aspergillus niger* or *Yarrowia lipolytica* | [211] |
| Orange peel | Orange peels as a substrate to produce pectinolytic, cellulolytic, and xylanolytic enzymes by (SSF) using fungi from the genera Aspergillus, Fusarium, and Penicillium | [66] |
| Grape by-products | Grape by-products have been used as a substrates for the production of hydrolytic enzymes such as cellulase and pectinase | [212] |
| Mango peel | Mango peels were used to produce lactic acid (up to 17.5g/L) and pectinase enzyme | [213] |
| Mango seed kernel | Mango seed kernels were used to produce α-amylase with *Fusarium soloni* | [214] |
| Pineapple peel | Pineapple peel can be used as a substrate for methane, ethanol, and hydrogen generation by *S. cerevisiae* and *Enterobacter aerogenes* | [8] |
| Pineapple peel | Pineapple peels have been anaerobically digested to yield biogas in the form of methane | [215] |
| Pineapple and orange peel | Bioethanol is produced from fruit peels of pineapple, orange, and sweet lime using *S. cerevisiae* | [216] |
| Papaya seed | Papaya seeds are used as biosorbents to remove heavy metals such as lead and cadmium | [217] |

## 9. Summary and Research Needs

This review presents important information on fruit processing and by-product utilization. Several previous studies have confirmed that depending on the type of fruit, variety, and cultivation conditions, a loss of up to 60% occurs. Such a huge loss of fruit by-products has a significant negative implication for the economy, environment, and social well-being worldwide. Fruit by-products are good sources of nutrients and bioactive components, implying that they could have an important role in functional food development. These bioactive compounds have anticancer, antidiabetic, antimicrobial, antioxidative, and immune-modulatory effects, confirming their role in nutraceuticals. Furthermore, fruit processing and by-products are also used as substrates for the production of organic acids, essential oils, enzymes, fuel, biodegradable packaging, and preservatives. Given the significant importance of fruit-processing by-products in food insecurity alleviation,

health promotion, and environmental sustainability, further studies aiming at addressing this knowledge gap are greatly important.

**Author Contributions:** Conceptualization, E.T. and T.A.T.; methodology, E.T., T.A.T. and M.M.U.; writing—original draft preparation, E.T. and M.M.U.; writing—review and editing T.A.T., M.M.U., R.N., J.R.R., D.V.H. and T.A.; supervision, T.A.T., R.N., J.R.R., D.V.H. and T.A. All authors have read and agreed to the published version of the manuscript.

**Funding:** This research received no external funding.

**Institutional Review Board Statement:** Not applicable.

**Informed Consent Statement:** Not applicable.

**Data Availability Statement:** Not applicable.

**Conflicts of Interest:** The authors declare no conflict of interest.

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
