# Peer review of "Fruit By-Products and Their Industrial Applications for Nutritional Benefits and Health Promotion: A Comprehensive Review"

_sustainability, doi:10.3390/su15107840_

Round 1
Reviewer 1 Report
Dear Editor
The manuscript explored the fruit by-products and their industrial applications for nutritional benefits and health promotion: A comprehensive review. They showed that fruit by-products are a good source of nutrients and bioactive components. My overall evaluation of the manuscript is negative. There are a number of major revisions, formal and scientific aspects that should be addressed.
Little explanation has been given about the by-products of fruits and their uses. For example, it has been mentioned that banana peel has a good source of phenol compounds, but its use has not been mentioned. The reader does not realize the potential of using these by-products in the industry. In the results section about anti-cancer, anti-diabetic, anti-microbial, anti-oxidative, and immune-modulatory effects, confirming their role in nutraceuticals is explained, but the text does not mention which of these materials can be used for these applications. The current text is very incomplete and needs to provide the reader with a comprehensive explanation of the potential of these materials in industry and technology.
The article is not acceptable in its current form.
Reviewer 2 Report
The topic of the review is interesting since it is necessary to know the available alternatives to take advantage of the fruit by-products that are generated during the processing of these, and thus reduce the environmental impact generated. However, I consider it important to deepen precisely these alternatives, since these alternatives are named in a very general way, and it would be very interesting to know more information so that they could be applied to continue with the research and achieve the development of products that are commercially viable.
In lines, 192 to 194 is indicated the information that the readers found in Tables 1, 2, 3, and 4, for this reason, I considered that it is not necessary to repeat this information in lines 205 to 207, 214 to 214, 325 to 236. The same behavior occurs in tables 5, 6, and 7.
Lines 169. The information "Tables 5, 6 and 7 summarize the industrial applications of fruit by-products" is not related to the topic of item 4: nutritional potentials of fruit by-products. Review if it is ok to include this information in this item.
Line 239, in item 5 - current knowledge on the utilization of fruit by-products, is poor lacking more depth on the topic, considering that this is the focus of the article. An analysis of the information shown in the tables should be included or more information should be provided to allow its application.
Reviewer 3 Report
Dear Authors,
1) Please revisit your abstract to provide a complete summary of your study, including the Objective of the study; Methodology; Findings; Conclusion; and Implication.
2) Research design and research approach need further elaboration.
3) Why you chose the mixed method approach? No explanation in detail.
4) Provide references from the literature on research methods.
5) How do you collect the data?
6) Please explain in the discussion section.
7) Summary and research need further explanation (ex: implication of the study).
Reviewer 4 Report
Dear authors,
This review article discusses the utilization of fruit by-products at the industrial level and their importance in nutritional and health aspects. Although, you have discussed the most important topics related to the title of the review article I would suggest a few additions more as below.
1. It will be much better if you can briefly explain/ introduce/ define the by-products of the fruits as a different section at the beginning. Here, you can use the pictures as well.
2. At the end of the introduction, can you please include the objective/s of this review article?
3. If you can include a separate section on the health aspects of fruit by-products then all your topics will be covered in the manuscript.
4. Brief explanations on a few selected processing methods of fruit by-products at the industrial level also will add value to your manuscript.
4. Finally, if possible please use a few diagrams/ pictures to show the industrial processing/ methods of selected by-products.
Thank you
Round 2
Reviewer 1 Report
The changes made are acceptable.
Reviewer 2 Report
This new version of the manuscript has been improved and is clearer and more in-depth on the issues associated with the health benefits and methods of extraction of bioactive compounds from fruit by-products.